# Leveraging Automated Unit Tests for Unsupervised Code Translation

**Baptiste Rozière**
Facebook AI Research
Paris-Dauphine University
broz@fb.com

**Jie M. Zhang**
University College London[†]
zhangjie@fb.com

**François Charton**
Facebook AI Research
fcharton@fb.com

**Mark Harman**
Facebook
markharman@fb.com

**Gabriel Synnaeve**
Facebook AI Research
gab@fb.com

**Guillaume Lample**
Facebook AI Research
glample@fb.com

## Abstract

With little to no parallel data available for programming languages, unsupervised methods are well-suited to source code translation. However, the majority of unsupervised machine translation approaches rely on back-translation, a method developed in the context of natural language translation and one that inherently involves training on noisy inputs. Unfortunately, source code is highly sensitive to small changes; a single token can result in compilation failures or erroneous programs, unlike natural languages where small inaccuracies may not change the meaning of a sentence. To address this issue, we propose to leverage an automated unit-testing system to filter out invalid translations, thereby creating a fully tested parallel corpus. We found that fine-tuning an unsupervised model with this filtered data set significantly reduces the noise in the translations so-generated, comfortably outperforming the state-of-the-art for all language pairs studied. In particular, for Java → Python and Python → C++ we outperform the best previous methods by more than 16% and 24% respectively, reducing the error rate by more than 35%.

## 1 Introduction

Ancient languages such as COBOL still underpin much of the financial industry and government services. Their outdated structures and thinning developer bases induce costs and severely slow down development, prompting businesses to modernize their codebases. For instance, the Commonwealth Bank of Australia spent around $750 million over 5 years to migrate its COBOL codebase to a more recent language. More generally, most large companies own code written in several programming languages, which can hinder interoperability and make programmers less efficient. Automatic translation systems could make codebase migrations faster and cheaper, and help programmers learn new languages or understand existing code. Systems to automatically translate between programming languages with approximately the same level of abstraction are called transpilers or source-to-source compilers. They need to be distinguished from compilers which translate source code to a lower-level language. The particularities of some languages allow the creation of very successful rule-based transpilers for a few language pairs (e.g. Java→Scala, CoffeeScript→JavaScript). Methods leveraging verified lifting (Kamil et al., 2016), which offer formal guarantees, can significantly speedup some pre-defined code fragments (Ahmad & Cheung, 2016; Ahmad et al., 2019).

However, source-to-source translation for arbitrary programming languages is still an open problem. Rule-based systems are commonly used, but they are never exhaustive due to the considerable number of translation rules that should be written to translate every function and object from every standard library. Unlike in natural languages, there is little to no parallel data available for source code, making it impossible to train standard machine translation models. Recently, TransCoder (Roziere et al., 2020) showed that unsupervised methods can be used to translate source code. However, it is

---

[†]Work done while at Facebook

trained without any supervised signal and only learns the semantics of tokens from their contexts. As shown in Figure 1, it can confuse tokens that have different semantics in different languages, for instance the float division in Python and integer division in C++ and Java which use the token / or more subtle operator priority differences (e.g. Java prioritizes == over & unlike Python). While small inaccuracies often merely hinder comprehension in natural languages, they often make the entire translation erroneous in the context of programming languages.

| Input function | TransCoder | TransCoder-ST |
|---|---|---|

```python
def is_odd(x):
  return x & 1 == 1
```

```java
static void printb(int x){
  while (x > 0){
    System.out.println(x % 2);
    x /= 2;
  }
}
```

```java
static String reverse(char[] str){
  Stack <Character> st = new Stack<>();
  for(int i = 0; i<str.length; i++)
    st.push(str[i]);
  for(int i = 0; i<str.length; i++){
    str[i] = st.peek();
    st.pop();
  }
  return String.valueOf(str);
}
```

```java
static boolean isOdd(int x) {
  return x & 1 == 1;
}
```

```python
def printb(x):
  while x > 0:
    print(x % 2)
    x /= 2
```

```python
def reverse(str):
  st = Stack()
  for i in range(len(str)):
    st.push(str[i])
  for i in range(len(str)):
    str[i] = st.pop()
    st.push(str[i])
  return str
```

```java
static boolean isOdd(int x) {
  return ( x & 1 ) == 1;
}
```

```python
def printb(x):
  while x > 0:
    print(x % 2)
    x //= 2
```

```python
def reverse(data):
  st = []
  for c in data :
    st.append(c)
  for i in range(len(data)):
    data[i] = st[-1]
    st.pop()
  return ''.join(data)
```

Figure 1: **Improvements over TransCoder.** The first function returns whether an input integer is odd and is translated from Python to Java. The translation of TransCoder does not compile because the == operator has precedence over & in Java, and parentheses are required unlike in Python. The second example is a function that prints an integer in base two, which is translated from Java to Python. TransCoder translates does not modify the expression x/=2, even though it corresponds to the integer division in Java and to the float division in Python. In the third example, a function reversing a char array, TransCoder does not manage to translate the Java Stack object into the right Python object and uses the unsafe str parameter name. In all three cases, TransCoder-ST manages to leverage the semantics contained in unit tests to translate the function correctly.

TransCoder leverages back-translation (Sennrich et al., 2015), an effective data-augmentation scheme where the model translates source sequences to generate training data for the target-to-source direction, and vice versa. Although being highly effective in low-resource translation, back-translation also has issues, as the model is trained on potentially invalid input-output pairs. Neural machine translation models being highly sensitive to input noise (Belinkov & Bisk, 2018; Khayrallah & Koehn, 2018), this can severely deteriorate the performance. Fortunately, many programming languages come with relatively mature tools and technologies for automated test data generation. In this paper, we propose to leverage these tools to guide the translation process, weeding out unsuccessful translations, thereby increasing the overall confidence in the machine translation process.

The topic of automated test data generation has been active for over three decades in the software engineering research community (Myers, 1979; Miller & Spooner, 1976). There are now many existing mature tools for test data generation, both open source research tools (Fraser & Arcuri, 2011; Lakhotia et al., 2013; Cadar et al., 2008), and production testing systems (Alshahwan et al., 2018; Tillmann et al., 2014). Because of its pivotal impact on practical software engineering, automated testing remains a highly active research area (Anand et al., 2013), with the result that future automated testing advances will lead to ongoing improvement in automated translation.

We use one such open source automated test generation tool, EvoSuite (Fraser & Arcuri, 2011), in this paper. EvoSuite is a well-established test generation tool for Java which uses coverage metrics (Chekam et al., 2017) and mutation scores (Jia & Harman, 2011) to generate high-quality tests. It has been widely used in the Software Testing research literature for test data generation although it has not, hitherto, been used as part of an automated code translation approach, the topic of the present paper.

More generally, software testing tools have been largely ignored by the machine learning community (Zhang et al., 2020). In this paper, we propose to use automatically created unit tests to guide unsupervised translation models for programming languages. More precisely, we create unit tests automatically for a large number of functions from the source dataset. Since the unit tests are composed of simple inputs and asserts, they can easily be translated to semantically equivalent tests in the target languages using simple scripts. Using our unit-tests and a pre-trained unsupervised

translation model, we create parallel datasets by translating functions and selecting the translations that have the same semantics as the original function for the tested inputs. Overall, we make the following contributions:

- We introduce a novel approach, TransCoder-ST (for Self-Trained), that leverages an automated unit test generation pipeline to filter out invalid translations and reduce the noise coming from the back-translation process in unsupervised machine translation.

- We present two implementations of this approach (online and offline), and show that it significantly outperforms the previous state of the art in code translation on all the language pairs we considered. In particular, we improve the state of the art for translating between Java, Python and C++ by an average of 12.6% Computational Accuracy (CA@1), corresponding to an average relative improvement of 25.5%. For Python $\to$ C++, we improve the CA@1 by 24%, reducing the error rate by 35.7% compared to previous models.

- We generate multilingual unit tests for hundreds of thousands of Java functions and create a large parallel dataset of 135,000 parallel functions between Java, Python, and C++.

- Our method is completely unsupervised and could easily be generalized to other programming languages and unit test creation tools.

## 2 RELATED WORK

**Unit Test Generation.** Software testing is challenging due to the large number of possibilities to be tested, and the inherent cost of covering reasonable representative sample (Myers, 1979). When test design is performed by humans, the cost can be prohibitive. To reduce such cost, much research over the last three decades has focused on automating the process of test generation (Anand et al., 2013). Although automated test generation has been studied since the mid-1970s (Miller & Spooner, 1976), it was only in the last decade that industrial-strength tools have become widely available. There are now several test data generation tools for languages, including C (Cadar et al., 2008; Lakhotia et al., 2013) and Java (Fraser & Arcuri, 2011). Popular test data generation techniques include symbolic execution of the code (Cadar & Sen, 2013), dynamic execution guided by a fitness function (Harman et al., 2015), and hybrids of these two techniques (Baars et al., 2011). Recently, neural networks have also been used successfully to generate unit tests (Tufano et al., 2020).

One of the most well-established and widely-used open source tools for test data generation is the EvoSuite system (Fraser & Arcuri, 2011). EvoSuite uses search based software engineering (SBSE) (Harman et al., 2012) to generate test cases. Like all SBSE techniques, EvoSuite is guided by fitness functions, in this case aimed at capturing the test suite's coverage and mutation score of the code being tested. We use EvoSuite in our work for three reasons: it is publicly available in open source (thereby facilitating replication), it is under current active development (thereby supporting future work), and it is widely used by other researchers (thereby enabling interoperability). The test framework can be considered as a parameter in our overall approach and could be substituted with another.

In order to assess the effectiveness of the test suites generated, we use mutation testing, a topic also widely-studied since the 1970s (DeMillo et al., 1978). A mutant is a version of the program into which a fault is deliberately inserted, thereby assessing the test suite's fault detection ability (Jia & Harman, 2011; Papadakis et al., 2019). For a given set of mutants and a test suite, the mutation score is defined to be the proportion of mutants for which the test suite distinguishes the behavior of the mutant from that of the original program. The mutation score is thus a proxy for the fault-revealing power of the test suite on a set of simulated faults (the mutants). Mutation scores have been empirically demonstrated to be correlated to real fault revelation (Chekam et al., 2017), motivating our adoption of this approach.

**Machine Learning for Programming Languages.** In recent years, deep learning methods have been used to tackle various tasks in software engineering, with a particular interest in bug detection and repair (Wang et al., 2018; Chen et al., 2019; Allamanis et al., 2018; Tarlow et al., 2020; Murali et al., 2021; Dinella et al., 2020; Yasunaga & Liang, 2020; Tufano et al., 2019; Drain et al., 2021) and code completion (Li et al., 2018; Liu et al., 2020; Kim et al., 2021; Svyatkovskiy et al., 2021). Unsupervised pre-training methods for code based on BERT (Kanade et al., 2020; Feng et al., 2020),

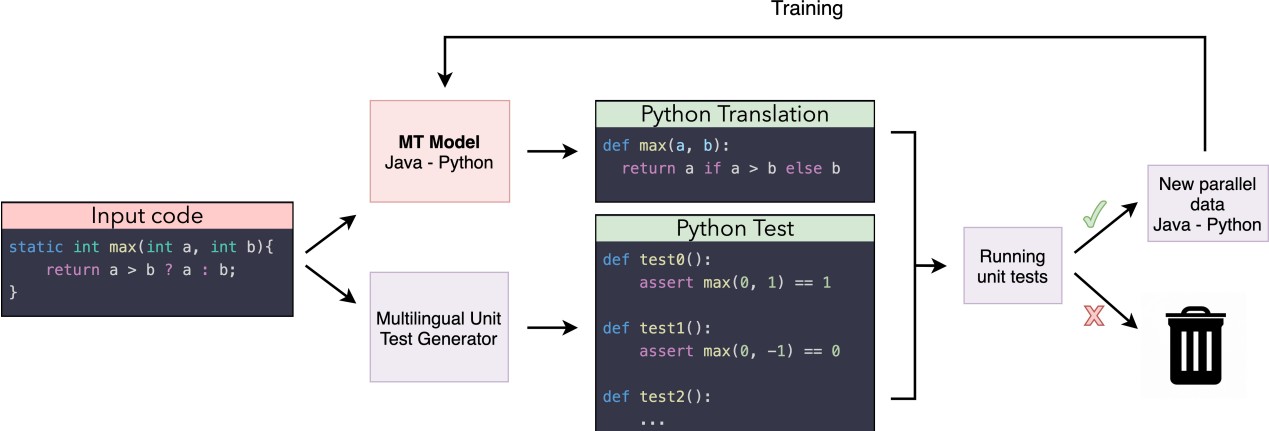

Figure 2: **Our iterative self-training method.** Using EvoSuite, we generate unit tests in Java, Python and C++ corresponding to several input Java functions. With a machine translation model (e.g. TransCoder), we generate several candidate translations of the the Java function in Python and C++. Generated translations that pass the unit tests are used to create a parallel dataset on which we fine-tune the model. Discarding translations that fail the unit tests reduces the noise of data coming from the back-translation process, and significantly improves the overall performance of the model.

BART (Ahmad et al., 2021) or other objectives tailored to source code (Guo et al., 2020; Roziere et al., 2021) have shown strong results on benchmarks such as CodeXGLUE (Lu et al., 2021).

Recently, Hendrycks et al. (2021) evaluated the competence of several language models for solving coding challenges. Chen et al. (2021) trained a a large model to generate programs from docstrings and are able to solve 28.8% of the problems in their HumanEval dataset. Austin et al. (2021) also evaluated the capabilities of large language models for generating code solving problem statements written in natural language. The goal of code translation is also to generate code solving a specific problem, but the input (code written in a different language) is more precise and often more concise.

**Translation of Programming Languages**   Several studies used statistical methods to translate between programming languages. Early methods extracted parallel datasets and trained phrase-based models to translate between C# and Java (Nguyen et al., 2013; Karaivanov et al., 2014) or from Python 2 to Python 3 (Aggarwal et al., 2015). Later, Chen et al. (2018) proposed a tree-to-tree neural network to translate between CoffeeScript and JavaScript and between C# and Java using the dataset created by Nguyen et al. (2013). However, these approaches are limited to a few language pairs for which small parallel datasets were created manually (e.g. C#-Java) or can be created with rule-based tools (e.g. Python 2-Python 3 and CoffeeScript-JavaScript).

Instead, Roziere et al. (2020) proposed TransCoder, an unsupervised model that leverages the principles of unsupervised machine translation (Lample et al., 2018), to translate between Python, Java and C++. They showed that their method outperforms well-established rule-based baselines, does not require any parallel data or expert knowledge, and can easily be generalized to other languages. They pre-trained their model with the Masked Language Modeling (MLM) objective of Devlin et al. (2018), and trained it with the denoising auto-encoding (DAE) (Vincent et al., 2008) and the back-translation (BT) (Sennrich et al., 2015) objectives. Later, Roziere et al. (2021) showed that augmenting MLM with a deobfuscation objective (dubbed DOBF) can substantially improve the performance of TransCoder. In the rest of the paper, we will refer to their model as DOBF.

Even though unsupervised methods can be trained on large amounts of data, they sometimes lack the signal needed to differentiate between semantically different tokens that often occur in similar contexts (see Figure 1). There is a need for a method providing supervised signal directly related to the semantics of the code without manually crafted parallel datasets.

## 3 METHOD

### 3.1 PARALLEL DATA CREATION

**Parallel unit test generation:**   We use EvoSuite to automatically generate unit tests for Java functions. EvoSuite is a well-established open source tool for automated test generation in Java, which is

still under active development and frequently used. It is designed for Java programs but its search-based technique is general and could be used for any programming language. Unit tests can be thought of as lists of inputs and asserts testing the semantics of a program (e.g., the output of the function, the side effects on its arguments such as sorting the input list). EvoSuite uses evolutionary methods to derive tests that maximize criteria such as code coverage or mutation score. During its search, each candidate solution in EvoSuite is a test input. The candidate inputs are evolved using crossover and mutation, and filtered by a fitness function (e.g., mutation score). With each generation the fitness improves until it reaches a plateau or the budget is exhausted. The final test inputs are wrapped up as test cases. Each program is associated to a test suite containing a series of test cases. Figure 3 shows an example of a test case generated by EvoSuite.

| Java function | A generated unit test |
|---|---|

```java
public class CLAMP_CLASS{
    public static double clamp(
    double a, double min, double max){
        return a<min?min:(a>max?max:a);
    }
}
```

```java
@Test(timeout = 4000)
  public void test0()  throws Throwable  {
      double double0 = Example.clamp(
      742.0, 0.0, 0.0);
      assertEquals(0.0, double0, 0.01);
  }
```

Figure 3: **A unit test generated by EvoSuite.** The Java function clamps the given value $a$ between the given $min$ and $max$. This test case is not sufficient to test the semantics of the function thoroughly but could be part of a suitable test suite. See Figure 5 in the Appendix for a generated test suite with a high mutation score.

**Parallel test suites selection:**   Some test suites created by EvoSuite only cover a few parts of the semantics of functions. We only trust the translations verified by test suites which examine the function semantics thoroughly. We use the mutation score, which is the most effective test assessment metric in the literature (Jia & Harman, 2011), to pick out these test suites. The mutation score is computed through mutation testing, in which mutants (i.e., program variants with syntactic changes) are generated from the original program based on a set of transformation rules (more details in Appendix A.2). A mutant is said to be killed if at least one test from the test suite has different results on the mutant and the original program. Otherwise, the mutant is said to survive. The mutation score is the ratio of killed mutants. A test suite with a higher mutation score checks the code semantics more thoroughly. We adopt a strict strategy in test suite selection: we keep only the Unit test suites with a mutation score larger than 90% for building the parallel dataset.

**Parallel dataset building:**   The generated test suites can be used to test the semantics of programs written in any programming language as long as there is a clear mapping between the types of the output and parameters in the original language and the language of the translated unit tests. We transform the generated Java tests into C++ and Python tests with identical inputs and expected outputs and side effects (i.e., assertions). In practice, we selected the Java functions which can be compiled and run in isolation and with simple output and parameter types. These types are the Java primitive types (e.g. `int`, `long`, `bool`, `float`...), standard data types (e.g. `Integer`, `Double`, `String`...), array and `List` or `ArrayList` types of elements of supported types (e.g. `double[]`, `List<Integer>`...).

We use the best unsupervised translation models available for Java to Python and Java to C++ translation, namely TransCoder (Roziere et al., 2020) for Java to C++ and DOBF (Roziere et al., 2021) for Java to Python. For each Java function, we generate 20 Python and C++ translations with beam search and select the first element in the beam that passes the unit tests. The created tests are executed against the translated functions. If all the tests pass, the Python and C++ functions have the same semantics assessed by the generated tests. Our method is illustrated in Figure 2.

## 3.2   TRAINING METHOD

Our parallel data generation method relies on a pre-existing model to translate from Java to Python and C++. There is little parallel data for these tasks and the best performing published models are unsupervised. TransCoder (Roziere et al., 2020) is trained using the MLM, denoising and back-translation objectives and is able to translate between Java, C++ and Python. DOBF (Roziere et al., 2021) provides clear improvements over TransCoder for translating between Java and Python but was not trained on C++. Therefore, we use DOBF to translate from Java to Python and TransCoder

Table 1: **Size of the parallel datasets generated offline at each iteration.**

| Languages | First iteration | Second iteration | Third iteration | Fourth iteration |
|---|---|---|---|---|
| Java ↔ C++ | 27,875 | 37,769 | 47,729 | 60,495 |
| Java ↔ Python | 33,496 | 43,194 | 43,956 | 45,311 |
| C++ ↔ Python | 14,935 | 21,026 | 27,080 | 32,869 |

to translate from Java to C++. When fine-tuning, we also reload these models. For DOBF, we initialize the C++ language embeddings with those of Java.

The parallel examples we generate can be used to improve the performance of pre-existing translation models. Since the number of examples we generate also depends on the performance of the translation model, it creates a positive feedback loop where improving the model allows to improve the parallel dataset which in turn can be used to improve the model again. We propose offline and online approaches to use our method to maximize the unsupervised translation performance.

**Offline training.** With the offline training method, we use the method described in Section 3.1 to create parallel Java ↔ Python, Java ↔ C++ and Python ↔ C++ datasets using every input Java function we selected. For the first iteration, we fine-tune the model on these parallel examples until convergence. We can iterate this process by selecting the best checkpoints for Java → Python and Java → C++ using the validation dataset and using them to generate new parallel datasets, which can in turn be used to train a better model. We iterate this process until convergence, i.e. when we see no significant improvements on the validation set.

**Online training.** With the online method, we create parallel examples on the fly while training the model. Compared to the offline method, it allows to always use the last model to generate new examples and it is much more convenient to automate. However, this process can be unstable if done naively. For instance, the model can start over-fitting only a few examples and stop generating anything that passes the unit tests for any other example. In order to stabilize the training, we follow Likhomanenko et al. (2020) and implement a cache mechanism storing the previous examples that passed the unit tests. At each step, the model can either train on parallel functions sampled from the cache or create new parallel functions to add to the cache. When an example is sampled, we remove it from the cache with a given probability. The online training allows the model to always benefit from the performance of the latest model and the cache mechanism ensures that the model does not forget the correct examples that it was able to generate at previous time steps.

## 3.3 EVALUATION

In the context of natural languages, machine translation models are generally benchmarked against a reference solution using the BLEU score (Koehn, 2009; Bahdanau et al., 2015; Vaswani et al., 2017). Early studies on source code translation used the same metric to evaluate the quality of the generated functions (Nguyen et al., 2013; Karaivanov et al., 2014; Aggarwal et al., 2015; Miceli-Barone & Sennrich, 2017), or the exact match score which requires the translation to be exactly equal to the ground truth (Chen et al., 2018). However, these metrics fail to capture the semantics of the code and typically correlate poorly with the correctness of the generated function, prompting the use of new metrics checking if the generated solution passes series of test cases (Kulal et al., 2019; Roziere et al., 2020; Hendrycks et al., 2021; Chen et al., 2021; Drain et al., 2021).

We evaluate our models on the full validation and test sets of TransCoder. It contains a few hundreds of parallel functions extracted from GeeksforGeeks along with associated unit tests. As our TransCoder and DOBF baselines, we evaluate our models with the CA@N metric, which checks if any of the top-N solutions proposed by the model passes all the corresponding unit tests. This metric can be computed independently of the beam size (as long as the beam size is greater or equal to N).

## 4 EXPERIMENTS

### 4.1 TRAINING DETAILS

**Model architecture.** We use a sequence-to-sequence model with attention composed of an encoder and a decoder model with a transformer architecture (Vaswani et al., 2017). In order to pro-

Table 2: **Computational accuracy scores for our methods and baselines.** We show the CA@1 metric computed with beam size 10. For the baselines, we ran the evaluations again and reported the best result between those reported in the original paper and those we obtained. Both the offline and online self-training methods lead to significant improvements over our baselines for every language pair and direction. Online self-training outperforms offline self-training, even after several iterations.

|  | C++ → Ja | C++ → Py | Ja → C++ | Ja → Py | Py → C++ | Py → Ja | AVG |
|---|---|---|---|---|---|---|---|
| TransCoder | 65.1% | 47.1% | 79.8% | 49.0% | 32.6% | 36.6% | 51.7% |
| DOBF | - | - | - | 52.7% | - | 45.7% | - |
| Offline ST 1 | 65.5% | 56.2% | 81.6% | 61.8% | 46.8% | 55.1% | 61.1% |
| Offline ST 2 | 65.5% | 58.3% | 83.7% | 63.3% | 46.4% | 52.2% | 61.6% |
| Offline ST 3 | 66.5% | 56.2% | **85.2%** | 66.3% | 48.1% | 56.6% | 63.1% |
| Offline ST 4 | 65.3% | 48.2% | 81.1% | 58.1% | 48.9% | 54.7% | 59.4% |
| Online ST | **68.0%** | **61.3%** | 84.6% | **68.9%** | **56.7%** | **58.2%** | **66.3%** |

vide fair comparisons, we use the exact same architecture as TransCoder: an encoder and a decoder of 6 layers each, a hidden dimension of 1024 and 8 attention heads. We limit the size of the input to 512 tokens. Roziere et al. (2021) train models with two different architectures. For Java ↔ Python, we compare ourselves to the version of DOBF using the same architecture as TransCoder. We initialize our models with either the best TransCoder checkpoint for Java → C++ or the best DOBF checkpoint for Java → Python with C++ language embeddings initialized with those of Java.

**Datasets.** As TransCoder and DOBF, we use the GitHub public dataset available on Google Big-Query filtered to keep only projects with open-source licenses[1]. As our unit test creation tool can only be used on Java code, we only use the Java files and we select only the functions that can be compiled in isolation. We obtain a dataset containing 333,542 Java functions. We run EvoSuite with a budget of 20 seconds and a criterion including the line, branch, cbranch and output coverages, as well as the weak and strong mutation scores. We set the maximum absolute value of integers that can be generated as an input to $\sqrt{2^{31}-1}$ to limit the number of overflows. We manage to obtain high-quality (mutation score > 0.9 and at least two asserts) test cases for 103,488 functions. See Figures 3 and 5, 6 in the appendix for examples of selected and filtered out test suites.

**Training details.** During the training, we alternate between batches for every source and target language so that language pairs for which we managed to create more parallel examples are not overrepresented in our training batches. For the online version, we set a cache warm-up parameter to ensure that we always generate new parallel examples if there are less than 500 examples in the cache for any language pair. Otherwise, we sample from the cache with probability 0.5, or generate new examples, train on them once and put them in the cache also with probability 0.5. The sampled elements are removed from the cache with probability 0.3, so that each element we create is trained on about 4 times in average before being removed from the cache. We initialize the cache with parallel examples created offline.

During beam decoding, we compute the score of generated sequences by dividing the sum of token log-probabilities by $l^\alpha$ where $l$ is the sequence length. We found that taking $\alpha = 0.5$ (and penalizing long generations) leads to the best performance on the validation set.

## 4.2 RESULTS AND DISCUSSION

**Results.** In Tables 2 and 3, we compare the results of our offline and online training methods with those of TransCoder and DOBF. DOBF outperforms TransCoder for the Java ↔ Python pair. We compare our models against the best baseline for each language pair and direction.

Training on the generated parallel examples brings substantial improvements for every language pair, direction, and metric. Offline training already provides clear improvements over the baseline after one iteration. The computational accuracy (CA@1) computed with beam size 10 is higher for every direction and it is substantially higher for the language pairs involving Python. It allows to reduce the error rate of the best baseline by 25.5% for Java → Python. In average, it increases the CA@1 by 7.4% over the best previous models, and reduces the error rate by 16.6%. In the

---

[1]We select the open-source licenses: 'apache-2.0', 'mit', 'gpl-2.0', 'gpl-3.0', 'bsd-2-clause', 'bsd-3-clause'

Table 3: **CA@n metric for several beam sizes averaged on all language pairs.** The value k corresponds to the beam size. For instance, CA@1 k=10 means that we use beam decoding to generate 10 translations, and select the one with the highest score. The best baseline corresponds to taking the best model between TransCoder and DOBF for every language pair and direction. The error rate reduction of the offline and online self-training methods over the best baseline are high ($> 20\%$) across all CA@N metrics and beam sizes.

|  | CA@1 k=1 | CA@1 k=10 | CA@1 k=20 | CA@10 k=10 | CA@20 k=20 |
|---|---|---|---|---|---|
| Best baseline | 52.2% | 53.7% | 53.4% | 67.3% | 70.5% |
| Offline ST 1 | 60.8% | 61.1% | 61.1% | 72.9% | 75.3% |
| Offline ST 2 | 61.4% | 61.6% | 61.4% | 73.3% | 75.8% |
| Offline ST 3 | 61.7% | 63.1% | 63.0% | 73.3% | 75.8% |
| Offline ST 4 | 58.5% | 59.4% | 59.2% | 70.8% | 73.6% |
| Online ST | **64.7%** | **66.3%** | **66.3%** | **75.4%** | **77.2%** |

Table 4: **Ablation study.** We show the CA@1 metric computed with greedy decoding at evaluation time except for the last line where the beam size is set to 10. We evaluate models trained with no cache system, without initializing the cache (with or without selecting the tests with a minimum mutation score of 0.9), and a beam size of 1 when generating examples. We also compare the CA@1 score of our full model when evaluating with greedy decoding and with beam size 10. Using a pre-filled cache and selecting only the tests with a high mutation score lead to substantially better performance, although these steps are not necessary to outperform our baseline. The online method already performs well with greedy decoding at generation time, but generating with beam size 20 further improves the results.

|  | C++ $\rightarrow$ Ja | C++ $\rightarrow$ Py | Ja $\rightarrow$ C++ | Ja $\rightarrow$ Py | Py $\rightarrow$ C++ | Py $\rightarrow$ Ja | AVG |
|---|---|---|---|---|---|---|---|
| No cache | 66.5% | 52.7% | 83.7% | 60.3% | 41.2% | 51.8% | 59.4% |
| Cache not initialized | 64.9% | 51.6% | 82.4% | 62.4% | 46.6% | 52.6% | 60.1% |
| + No min mut. score | 64.0% | 50.1% | 82.6% | 60.9% | 47.4% | 47.0% | 58.7% |
| ST greedy decoding | 65.9% | 54.2% | 82.2% | 60.9% | 56.2% | 56.6% | 62.7% |
| Full model (ST beam 20) | 66.7% | 61.1% | 84.1% | 67.8% | 52.2% | 56.7% | 64.7% |
| + Eval beam 10 | **68.0%** | **61.3%** | **84.6%** | **68.9%** | **56.7%** | **58.2%** | **66.3%** |

two next iterations, the model is trained on significantly more examples (see Table 1). It results in average improvements of 2% points between the first and third iteration. Although the model for the fourth iteration is trained on more parallel samples, its performance on the test set of TransCoder is actually worse than after the third iteration. After three iterations, the model learned to generate more samples that pass the unit tests but some of them are actually incompatible with the types of translations expected by TransCoder (e.g. example with overflows in Figure 4), causing the computational accuracy score to go down.

The online self-training method provides further improvements over training on the pseudo-labeled examples offline. It outperforms every other method in every case except the third iteration of offline training for Java $\rightarrow$ C++. In average, this model outperforms the baseline by 12.6% points, corresponding to an error rate reduction of 25.5%. For Python $\rightarrow$ C++, it improves previous performance by more than 24% points, which corresponds to reducing the error rate by 35.7%. Examples of avoided errors can be found in Figure 1 and Appendix B. Overall, all our models significantly improve previous results. As shown in Table 3, these improvements are stable across several beam sizes and CA@n metrics. The CA@20 metric shows that the number of examples for which none of the 20 elements in the beam are correct is reduced by more than 22% with online self-training. It indicates that, even though we train only on the output of the model, our method does much more than reordering the elements in the beam and allows the model to find correct solutions that were not assigned a high probability by the baseline model. See Table 6 in the appendix for more results.

**Ablation study.** The results of our ablation study are shown in Table 4. Training online with no cache makes the training much less stable. The model improves at the beginning of training and we can select a few checkpoints where it performs well, but it ends up over-fitting a few examples it generated and the performance drops after a few epochs. Starting with an empty cache slows down the training and hinders generalization, leading to a clear drop in performance. We also try removing the minimum mutation score requirement for the model with no initial cache, which leads to even lower scores as the model is trained partly on lower-quality parallel data.

All these models were trained using a self-training beam size of 20 when generating new examples. Training with greedy decoding is much faster since computing the results for all the 20 elements

of the beam is costly. However, generating new examples with greedy decoding leads to a loss of about two percentage points in average compared to our full model using beams of size 20. It shows that initializing the cache of the model with beam size 20 is not sufficient and creating new examples with beam search is necessary to reach our best performance. Our full model provides some improvements over the ablated versions for every language pair and direction, except over the model trained with greedy decoding for Python → C++ translation. Evaluating with beam size 10 (still returning only the first element) leads to some improvements for every language pair.

**Limitations.** We found that the unit tests we create with this method are sometimes incompatible with those of the test set of TransCoder, and that the capacity of a model to generate functions that pass these unit tests is not perfectly correlated to its score on the test set. It raises the deeper issue of defining what constitutes a correct translation. For instance, most programmers would translate a factorial function implemented with `long` integers into a factorial function implemented with Python's integer type. However, these functions are not semantically equivalent since the Java implementation would return a negative number for the input `21` due to integer overflow while the Python implementation would return 21! correctly. The human developers who wrote the parallel functions in the test set of TransCoder often assumed that these functions would only be used on a limited domain where no overflow occurs (see Figure 4). However, the test cases of EvoSuite and TransCoder are not limited to this domain and they sometimes assert different semantics. By using the test suites from EvoSuite as source of truth, we sometimes train the model to generate translations that are more rigorous but also less natural.

| Input Java function | Gold translation | Translation passing multilingual tests |
|---|---|---|

```
static int factorial(int n){      def factorial(n):              def factorial(n):
    if (n < 2) return 1;              if n < 2:                      n = np.int32(n)
    return n * factorial(n - 1);          return 1                  if n < 2:
}                                     return n * factorial(n-1)          return np.int32(1)
                                                                     return n * factorial(n - 1)
```

Figure 4: **Example of disagreement between our multilingual tests and the test set of TransCoder.** The gold translation is only equivalent to the input Java function on a small domain where there is no integer overflow and does not pass our unit tests. The version that passes the unit tests casts uses the `np.int32` type, reproducing the behaviour of the original Java code but causing it to fail some of the unit tests of TransCoder.

## 5 CONCLUSION

In this paper, we introduced a novel method to grow a parallel corpus for automated code translation, from completely monolingual data. We leverage multilingual unit tests to filter good pseudo-labels, improving the model, and in turn the candidate translations. We show that both offline and online methods substantially improve the state of the art in unsupervised code translation, with an average improvement of 12.6% points in computational accuracy, and up to 24% points for Python → C++, corresponding to translation error rate reductions of 25.5% and 35.7% respectively, without using any unit test generation tool for Python and C++ (exclusively for Java).

Our method would automatically gain from improvements of automatic unit test generation tools. We could also increase the size of the dataset we generate by using test creation tools written for other languages in addition to Java, or by generating tests with EvoSuite on translated examples. Similarly, we could also extract the semantics of human-written unit tests found in open-source projects to obtain larger, and possibly higher-quality datasets. In this paper, we focused on translation correctness and our parallel example validation criterion was only based on semantics. It could be supplemented with other requirements, such as a specific code formatting or the output of linters to generate code verifying arbitrary criteria. Finally, the approach presented in this paper could easily be transferred to natural languages. Although there is no concept of unit tests in natural language, traditional grammar and syntax checkers could be used to filter out some incorrect generations, and reduce the noise coming from the back-translation process. Neural machine translation systems being highly sensitive to noise coming from parallel data (Belinkov & Bisk, 2018; Khayrallah & Koehn, 2018), this may improve the performance in low-resource machine translation significantly.

## REPRODUCIBILITY

We made sure to use the same architecture and framework as previous works in source code translation so that our results are comparable (see Section 4.1). We submit our code with this submission, along with a ReadMe file detailing clear steps to reproduce our results, including a script to set-up a suitable environment. We will open-source our code and release our trained models. Our models were trained using standard hardware (Tesla V100 GPUs) and libraries (e.g. Pytorch, Cuda) for machine-learning research.

## ETHICAL CONSIDERATIONS

In this paper, we improve source code translation methods. Our methods could facilitate codebase migrations and interoperability, encouraging companies to move away from ancient programming languages and making software developers more efficient. Although increased efficiency could reduce the number of developers needed to perform a task, its impact on the labor market is unclear as lower costs would also lower the bar for starting new projects and increase the demand for software engineers. Today, the demand for software engineering skills is high despite (or thanks to) the development of software (e.g. git, IDEs), programming languages (e.g. python), libraries (e.g. pytorch) and methodologies (e.g. continuous deployment) improving the efficiency of software developers, and we believe that the development of automatic translation tools would not drastically affect the prospects of software developers either. In the long term, the migration of codebases written in antiquated programming languages (e.g. COBOL) could negatively impact experts in those languages, who are in particularly high demand at the moment. However, it would also benefit society by facilitating debugging and updating software still used in most of our financial transactions and for many government services, and by increasing the demand for other software engineering skills.

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

# A MULTILINGUAL UNIT TESTS CREATION

## A.1 GENERATED UNIT TESTS

| Java function | Generated test suite |
|---|---|

```java
public static int pow (int b, int e) {
  int r = 1;
  while (e > 0) {
    if ((e & 1) == 1)
      r = r * b;
    b = b * b;
    e = e >> 1;
  }
  return r;
}
```

```java
public void test0()  throws Throwable  {
  int int0 = Example.pow((-1), (-1));
  assertEquals(1, int0);
}

public void test1()  throws Throwable  {
  int int0 = Example.pow(0, 1);
  assertEquals(0, int0);
}

public void test2()  throws Throwable  {
  int int0 = Example.pow((-13133), 2743);
  assertEquals((-1787379173), int0);
}

public void test3()  throws Throwable  {
  int int0 = Example.pow(1, 1);
  assertEquals(1, int0);
}
```

Figure 5: **A generated unit test suite with high mutation score.** The mutation score of this test suite is 95% and we selected it in our dataset for pseudo-labelling. The third test case (i.e. `test2`) may be too strict as it would make translations using the python int type fail the unit tests.

| Java function | Generated test suite |
|---|---|

```java
public static int sizeBits_cmd() {
  return 8;
}
```

```java
public void test0()  throws Throwable  {
  assertEquals(8, Example.sizeBits_cmd());
}
```

Figure 6: **A test suite with a good mutation score but only one assert.** Even though it contains only one test and one assert, this test suite tests the semantics of the function on the left properly since it only returns a constant and its mutation score is 100%. We found that test suites with good mutation scores and only one assert generally correspond to uninteresting input functions. Removing these functions and tests from our dataset for self labelling improves the performance of our model.

As discussed in Section 3.1, we only generate unit tests for static functions with selected return and parameter types. It makes it easy to map the types of inputs and outputs in Java to Python or C++ types in the translated unit tests. While most of the unit tests are translated correctly, the translation sometimes fails due to EvoSuite generating test cases expecting exceptions. Our analysis shows that it happens for about 5.6% of all tests and less than 2% of the tests with high mutation scores. In that case, the candidate translations cannot pass the translated tests and no parallel examples are created.

## A.2 MUTATION SCORE

In mutation testing, mutants are programs transformed from the original programs based on a series of syntactic transformation rules called mutation operators. Mutation testing consists in introducing minor syntactic faults on the code and running the tests against the mutated code. A strong test suite is expected to detect the code changes by having at least one test failing. Table 5 shows the examples of mutation operators adopted in EvoSuite when generating mutants (Fraser & Arcuri, 2015).

A mutant is said to be killed by a test case if the output of this test case on the mutant is different from its output on the original program (i.e., the test fails the mutant). Otherwise, the mutant is said to have survived. Figure 7 shows an example of a mutant generated by changing the $<$ in the return statement into $>$. The test with input (-800, -800, -1), as shown by Figure 3, does not kill this generated mutant, because its outputs on the original program and the mutant are the same.

Mutation score is considered as the most effective criteria in accessing the fault-revealing ability of test suites. Other criteria, such as code coverage, are weak: they check only whether the test

Table 5: **Examples of mutation operators in EvoSuite.**

| Mutation operator | Explanation |
| --- | --- |
| Delete call operator | Remove a method invocation |
| Delete field operator | Remove a field access and replaces it with a default value (0 / null) |
| Insert Unary Operator | Add 1 to, subtract 1 from, or negate a numerical value after it was loaded on the stack |
| Replace arithmetic operator | Replace an arithmetic operator in an expression with other operators. E.g., $+ \to -, * \to /$ |
| Replace constant operator | Replace constants with the special values -1, 0, +1 |
| Replace variable operator | Replace variables with other variables of the same type |

| Original Java function | Mutant |
| --- | --- |

```java
public class CLAMP_CLASS{
    public static double clamp(
    double a, double min, double max){
        return a<min?min:(a>max?max:a);
    }
}
```

```java
public class CLAMP_CLASS{
    public static double clamp(
    double a, double min, double max){
        return a>min?min:(a>max?max:a);
    }
}
```

Figure 7: **A mutant generated by the "Replace arithmetic operator" mutation in EvoSuite.** The $<$ operator in the return statement is replaced with $>$.

executes the code, but do not check whether the execution result is correct. A test suite without any assertions can achieve 100% code coverage, but could not detect any faults.

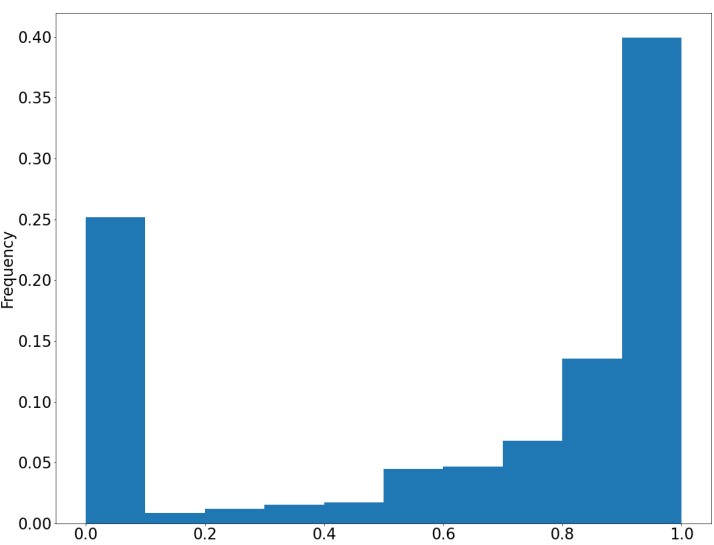

Figure 8: **Histogram of mutation scores for our generated unit tests.** We select about 40% of the unit tests with our threshold at 0.9. Many of the remaining unit tests have a mutation score of 0.

## B    TRANSLATION EXAMPLES

| Input Python function | TransCoder C++ translation | TransCoder-ST C++ translation |
|---|---|---|

```python
def rangeGCD(n, m):
    return n if (n==m) else 1
```

```cpp
int rangeGCD(int n, int m){
    return n == (n==m) ? 1: -1;
}
```

```cpp
int rangeGCD(int n, int m){
    return (n==m) ? n:1;
}
```

```python
def countSetBits(n):
  if(n == 0):
    return 0
  else:
    return(n & 1) \
        + countSetBits(n >> 1)
```

```cpp
int countSetBits(int n) {
  if((n == 0) ||(n == 1))
    return 0;
  else
    return(n & 1)
        + countSetBits(n >> 1);
}
```

```cpp
int countSetBits(int n) {
  if((n == 0))
    return 0;
  else
    return(n & 1)
        + countSetBits(n >> 1);
}
```

Figure 9: **Python to C++ translation examples.** TransCoder sometimes fails to capture the semantics of the incoming code and translates them to other expressions that could occur in similar contexts. Self-training helps the model to avoid such mistakes.

| Input Python function | TransCoder Java translation | TransCoder-ST Java translation |
|---|---|---|

```python
def cassini(n):
    return - 1 if(n & 1) else 1
```

```java
public static int cassini(int n) {
    return - 1 ==(n & 1) ? 1: - 1;
}
```

```java
public static int cassini(int n) {
    return(n & 1) != 0 ? - 1: 1;
}
```

```python
def turnOffK(n, k):
  if(k <= 0):
    return n
  return(n & ~(1 <<(k - 1)))
```

```java
static int turnOffK(int n, int k){
  if((k <= 0) (k > n))
    return n;
  return(n & ~(1<<(k - 1)));
}
```

```java
static int turnOffK(int n, int k){
  if((k <= 0))
    return n;
  return(n & ~(1<<(k - 1)));
}
```

```python
def isNumber(s):
  for i in range(len(s)):
    if s[i].isdigit() != True:
      return False
  return True
```

```java
static boolean isNumber(String s) {
  for(int i = 0; i<s.length(); i++){
    if(s.charAt(i) != ' ') {
      return false;
    }
  }
  return true;
}
```

```java
static boolean isNumber(String s){
  for(int i = 0; i<s.length(); i++){
    if(Character.isDigit(s.charAt(i))
        != true
      ){
      return false;
    }
  }
  return true;
}
```

Figure 10: **Python to Java translation examples.** Similarly to Python to C++, TransCoder often fails to get the right semantics, especially for conditions where it can hallucinate extra clauses or write incorrect comparisons. TransCoder-ST often solves these issues.

| Input Python function | TransCoder Java translation | TransCoder-ST Java translation |
|---|---|---|

```cpp
int cassini(int n) {
  return(n & 1) ? - 1: 1;
}
int power(int n) {
  if(n == 1) return 2;
  return 2 * power(n - 1);
}
```

```python
def cassini(n):
  return(n & 1)

def power(n):
  if n == 1:
    return 2
  return 2 ** power(n - 1)
```

```python
def cassini(n):
  return - 1 if(n & 1) != 0 else 1

def power(n):
  if n == 1:
    return 2
  return 2 * power(n - 1)
```

Figure 11: **C++ to Python translation examples.** For C++ to Python translation, many of the errors of TransCoder come from incorrectly translated conditions, wrong operators and badly translated functions. TransCoder-ST better comprehends the semantics of the code and is generally able to solve these issues.

| Input C++ function | TransCoder Java translation | TransCoder-ST Java translation |
|---|---|---|

```cpp
int minXOR(int arr[], int n){
  int min_xor = INT_MAX;
  for(int i = 0; i < n; i++)
    for(int j = i + 1; j < n; j++)
      min_xor = min(min_xor,
                    arr[i]^arr[j]);
  return min_xor;
}
```

```java
static int minXOR(int arr[], int n){
  int min_xor = Integer.MIN_VALUE;
  for(int i = 0; i < n; i++)
    for(int j = i + 1; j < n; j++)
      min_xor = Math.min(min_xor,
                         arr[i]^arr[j]);
  return min_xor;
}
```

```java
static int minXOR(int arr[], int n){
  int minXor = Integer.MAX_VALUE;
  for(int i = 0; i < n; i++)
    for(int j = i + 1; j < n; j++)
      minXor = Math.min(minXor,
                        arr[i]^arr[j]);
  return minXor;
}
```

```cpp
int findLongestConseqSubseq(
    int arr[], int n
  ){
  unordered_set<int> S;

  for(int i = 0; i < n; i++)
    S.insert(arr[i]);
  int ans = 0;
  for(int i = 0; i < n; i++) {
    if(S.find(arr[i]-1)==S.end()){
      int j = arr[i];
      while(S.find(j) != S.end())
        j++;
      ans = max(ans, j - arr[i]);
    }
  }
  return ans;
}
```

```java
public static
int findLongestConseqSubseq(
    int arr[], int n
  ){
  HashSet<Integer> S =
                new HashSet<Integer>();
  for(int i = 0; i < n; i++) {
    S.add(arr[i]);
  }
  int ans = 0;
  for(int i = 0; i < n; i++) {
    if(S.contains(arr[i] - 1)) {
      int j = arr[i];
      while(S.contains(j)) {
        j++;
      }
      ans = Math.max(ans, j - arr[i]);
    }
  }
  return ans;
}
```

```java
public static
int findLongestConseqSubseq(
    int arr[], int n
  ){
  HashSet<Integer> S = new HashSet<>();

  for(int i = 0; i < n; i++) {
    S.add(arr[i]);
  }
  int ans = 0;
  for(int i = 0; i < n; i++) {
    if(S.contains(arr[i] - 1)==false){
      int j = arr[i];
      while(S.contains(j)) {
        j++;
      }
      ans = Math.max(ans, j - arr[i]);
    }
  }
  return ans;
}
```

Figure 12: **C++ to Java translation examples.** In the first example computed the minimum XOR between two elements of an array, TransCoder erroneously translates INT_MAX into Integer.MIN_VALUE. This value is used in similar contexts (i.e. to compute a maximum instead of a minimum) but is inappropriate here. TransCoder-ST manages to correct this and outputs a function with the right semantics. In the second example, where the function computes the size of the largest subset of elements of the list that could form a sequence of consecutive integers, TransCoder-ST manages to translate the semantics of S.find(arr[i]-1)==S.end() appropriately while TransCoder translates it into its negation.

| Input Java function | TransCoder C++ translation | TransCoder-ST C++ translation |
|---|---|---|

```java
static boolean isCornerPresent(
            String str,
            String corner
  ) {
  int n = str.length();
  int cl = corner.length();
  if(n < cl) return false;
  return(str.substring(0,cl)
         .equals(corner)
      && str.substring(n - cl, n)
        .equals(corner));
}
...
PriorityQueue <Integer> Q =
    new PriorityQueue <Integer>();
...
```

```cpp
bool isCornerPresent(
    string str,
    string corner
  ) {
  int n = str.length();
  int cl = corner.length();
  if(n < cl) return false;
  return(str[0] == corner[0]
      && str[n - cl] == corner[n]);
}
...
priority_queue < int > Q;
...
```

```cpp
bool isCornerPresent(
    string str,
    string corner
  ) {
  size_t n = str.length();
  size_t cl = corner.length();
  if(n < cl) return false;
  return(str.substr(0, cl)==corner
      && str.substr(n-cl, n)==corner);
}
...
priority_queue<int,
               vector<int>,
               greater<int>
             > Q;
...
```

Figure 13: **Java to C++ translation examples.** In the first example, which returns whether a given string corner is present at the beginning and at the end of a string str, TransCoder completely fails to translate the last logical expression correctly while TransCoder-ST manages to translate the logic to get the right substrings and to return the right output. The second example is a line defining a priority queue extracted from the kthLargestSum function in the test set of TransCoder. The PriorityQueue object in Java returns the smallest elements first by default, while priority_queue in C++ returns the largest. TransCoder, which was not trained on any semantic signal, manages to instantiate a priority queue object but instantiates a max queue instead of a min queue. TransCoder-ST, which was trained with some supervised signal directly linked to the semantics of the code, manages to instantiate the right type of priority queue.

| Input Java function | TransCoder Python Translation | TransCoder-ST translation |
|---|---|---|

```java
static int divisorSum(int n) {
    int sum = 0;
    for(int i = 1; i <= n; ++i)
        sum +=(n / i) * i;
    return sum;
}
static
boolean check(int degree[], int n){
  int deg_sum = 0;
  for(int i = 0; i < n; i ++) {
    deg_sum += degree[i];
  }
  return(2*(n-1)==deg_sum);
}
static int decimalToBinary(int N){
  int B_Number = 0;
  int cnt = 0;
  while(N != 0) {
    int rem = N % 2;
    double c = Math.pow(10, cnt);
    B_Number += rem * c;
    N /= 2;
    cnt ++;
  }
  return B_Number;
}
```

```python
def divisor_sum(n):
  sum = 0
  for i in range(1 , n + 1):
    sum +=(n / i) ** i
  return sum

def check(degree, n):
  deg_sum = 0
  for i in range(n):
    deg_sum += degree[i]
  return(2**(n-1)==deg_sum)

def decimal_to_binary(N):
  B_Number = 0
  cnt = 0
  while N != 0:
    rem = N % 2
    c = pow(10, cnt)
    B_Number += rem * c
    N /= 2
    cnt += 1
  return B_Number
```

```python
def divisor_sum(n):
  sum = 0
  for i in range(1 , n + 1):
    sum +=(n // i) * i
  return sum

def check(degree, n):
  deg_sum = 0
  for i in range(n):
    deg_sum += degree[i]
  return(2*(n-1)==deg_sum)

def decimal_to_binary(N):
  B_number = 0
  cnt = 0
  while N != 0:
    rem = N % 2
    c = pow(10, cnt)
    B_number += rem * c
    N //= 2
    cnt += 1
  return B_number
```

Figure 14: **Translation examples for Java to Python.** When translating to Python, TransCoder was often failing to distinguish between float division (/) and integer division (//). It also often confuses other operations, for instance * and **. Training on self-created labels often solves this issue.

| Input Python function | TransCoder C++ translation | TransCoder-ST C++ translation |
|---|---|---|

```python
def transpose(A, B):
    for i in range(N):
        for j in range(N):
            B[i][j]=A[j][i]
```

```cpp
void transpose (
    const double* A ,
    double* B[N][N]
    ) {
  for(int i=0; i<N; i++)
    for(int j=0; j < N; j++)
        B[i][j]=A[j][i];
}
```

```cpp
template<typename T>
void transpose (T* A, T* B){
  for(int i=0; i<N ; i++){
    for(int j=0; j<N; j++){
      B[i][j] = A[j][i];
    }
  }
}
```

```python
def exponential(n, x):
    sum=1.0
    for i in range (n, 0, -1):
        sum=1+x*sum/i
    print("e^x =", sum)
```

```cpp
void exponential(int n, double x){
  double sum=1.0;
  for(int i=n; i>0; i--)
    sum=1+x*sum/i ;
  cout << "e^x =" << sum << endl ;
}
```

```cpp
template <typename T>
void exponential(int n, T x) {
  T sum=1.0;
  for(int i=n; i>0; i--)
    sum=1+x*sum/i;
  cout << "e^x =" << sum << endl;
}
```

Figure 15: **Our parallel unit tests lead to the generation of more general solutions using templates.** Solutions using templates can pass the unit tests for several parameter types, while guessing the wrong parameter type can lead to some errors. Solutions using templates succeed more often, are more likely to appear in the parallel data we generate and, as a result, in our model's generations. It leads to our model generating more templates (three times more often for our online model trained the longest).

## C EXTRA RESULTS

### C.1 BEAM REORDERING

We also evaluate a simpler method where we create unit tests for the Java functions in the test dataset and use them to reorder the elements of the beam at test time. We compute the results of the tests for every proposed C++ or Python translation and prioritize the elements that pass the unit tests.

As shown on Table 6, reordering the elements of the beam at test time when translating from Java leads only to small improvements compared to the best baseline (up to 1.7% CA@1 for Java → Python) and the scores of this method are far from those obtained when requiring any of the 10 element of the beam to be correct (i.e. CA@10). It can be explained by the fact that the tests generated by EvoSuite on these functions can have low mutation scores and be insufficient to thoroughly test the semantics of the functions. Moreover, the tests we create are sometimes incompatible with those of our test set (see Figure 4 for an example).

Table 6: **Extra results table.** We show the CA@1 metric computed with beam size 10 for our baselines, and our offline and online methods, the beam reordering, and a model trained from scratch with our dataset. Beam reordering leads only to small improvements compared to our offline and online self-training methods. Training on our generated parallel dataset from scratch leads to decent performances, but that are still below those of TransCoder and TransCoder-ST.

| | C++ → Ja | C++ → Py | Ja → C++ | Ja → Py | Py → C++ | Py → Ja | AVG |
|---|---|---|---|---|---|---|---|
| TransCoder | 65.1% | 47.1% | 79.8% | 49.0% | 32.6% | 36.6% | 51.7% |
| DOBF | - | - | - | 52.7% | - | 45.7% | - |
| Beam reordering | - | - | 80.3% | 54.4% | - | - | - |
| Offline ST scratch | 43.0% | 41.3% | 54.3% | 43.2% | 31.1% | 39.7% | 42.1% |
| Offline ST 1 | 65.5% | 56.2% | 81.6% | 61.8% | 46.8% | 55.1% | 61.1% |
| Offline ST 2 | 65.5% | 58.3% | 83.7% | 63.3% | 46.4% | 52.2% | 61.6% |
| Offline ST 3 | 66.5% | 56.2% | **85.2%** | 66.3% | 48.1% | 56.6% | 63.1% |
| Offline ST 4 | 65.3% | 48.2% | 81.1% | 58.1% | 48.9% | 54.7% | 59.4% |
| Online ST | **68.0%** | **61.3%** | 84.6% | **68.9%** | **56.7%** | **58.2%** | **66.3%** |

