# OpenReview forum: "Leveraging Automated Unit Tests for Unsupervised Code Translation"
_ICLR.cc/2022/Conference — ICLR 2022 Spotlight_

### Official Review · Reviewer_LnDd · 2021-11-02

**Correctness:** 4
**Technical Novelty And Significance:** 3
**Empirical Novelty And Significance:** 2
**Recommendation:** 8
**Confidence:** 4

**Main Review:**

The paper is well written and easy to understand. It provides a very nice overview of related work and the history of software testing. The core methods is very practical and easy to reproduce.

Using unit tests to improve quality of parallel datasets generated via backtranslation is not entirely new in the ML for software engineering domain. There were attempts to use it for bug patching applications, see for instance, DeepDebug (https://arxiv.org/abs/2105.09352) paper. It is also frequently used during inference, as a postprocessing step, and evaluation.

Nevertheless, a technique that would work in a multilingual code-to-code translation setting is non-trivial and novel. Generating a high-quality parallel corpus starting with monolingual data is non-trivial, it relies on a pre-trained translation model for the target task, and translating unit tests generated by EvoSuite from Java to target languages using heuristics and mapping of types. Using mutation scores to select test suites is also novel.

Authors claim that the unit tests suites can be used to test program semantics in any programming languages, provided there is a way to establish the mapping between types and parameters. While this is true in most cases, have you observed and analyzed some corner cases, for instance, when an object needs to be instantiated in the body of the unit test method prior to assertions. Instantiating such variables may rely on imported classes or require translating extensive surrounding code context other than the unit test itself. How often does it happen, are these cases ignored?

Starting with 333k Java functions, authors obtain 103k high-quality test case suites with EvoSuites which are then iteratively used to generate parallel training corpus. Have you analyzed the failed cases:
•	Does the model successfully generates data for small methods only (e.g. a single statement), and majority of failed cases are long methods? How does it handle test cases with multiple asserts?
•	When does EvoSuite fail to generate high-quality test suites? I wonder if EvoSuite scaffolding is an issue

Maybe I am missing something, but why would generated test set mismatch to TransCoder tests set be an issue? Are you generating tests sets for evaluation as well? Does this change the baseline (published) TransCoder computational accuracy numbers?

Minor:

For multiple published papers the authors cite corresponding arXiv pre-prints.
For instance:
1. Miltiadis Allamanis, Marc Brockschmidt, and M. Khademi. Learning to represent programs with graphs. ArXiv, abs/1711.00740, 2018 -> Published at ICLR 2018
2. Dzmitry Bahdanau, Kyunghyun Cho, and Yoshua Bengio. Neural machine translation by jointly learning to align and translate. -> published at ICLR 2015
… etc


"maximize a criteria such as code coverage or mutation score" -> "maximize criteria such as code coverage or mutation score"

“Otherwise, the mutant is said to be survived” -> “Otherwise, the mutant is said to have survived”

A1 caption (right), should perhaps be changed to “A generated unit test suite”

**Summary Of The Paper:**

The paper introduces a pipeline to generate parallel data corpus for code-to-code translation instrumented with unit tests, starting with monolingual data in Java programming language. This allows to improve data quality over noisy backtranslation approach and subsequently improve the model accuracy finetuned on these data. The approach is evaluated on the code-to-code translation against TransCoder baseline.

The authors also introduce two methods (online and offline) to train on these data, substantially improving upon the state of the art in unsupervised code translation, and reducing the error rate.

**Summary Of The Review:**

A well written paper that introduces a practical method to generate a supervised parallel corpus for code-to-code translation. The findings are  generally well supported by evaluation, but some additional tests could strengthen the paper.

---

> ### Author Response · Authors · 2021-11-16
> **Response to Reviewer LnDd**
>
> > There were attempts to use it for bug patching applications, see for instance, DeepDebug (https://arxiv.org/abs/2105.09352) paper. It is also frequently used during inference, as a postprocessing step, and evaluation.
>
> Thank you, we added a reference to this work in the related work section.
>
> > Authors claim that the unit tests suites can be used to test program semantics in any programming languages, provided there is a way to establish the mapping between types and parameters. While this is true in most cases, have you observed and analyzed some corner cases, for instance, when an object needs to be instantiated in the body of the unit test method prior to assertions. Instantiating such variables may rely on imported classes or require translating extensive surrounding code context other than the unit test itself. How often does it happen, are these cases ignored?
>
> We only generate tests for java methods with a restricted set of parameters and return types. Hence, none of our tests require to instantiate objects in the body of the unit test method and, in practice, we ignore those cases. It happens frequently and we keep only 333k functions from our original dataset containing more than 8M unique functions.
> This method learns general patterns which are useful for all types of functions, but, as you mentioned, it could be improved by translating extensive surrounding code context (i.e. a file/function and all its dependencies) which could be tested together.
>
> > Have you analyzed the failed cases: • Does the model successfully generates data for small methods only (e.g. a single statement), and majority of failed cases are long methods?
>
> The model successfully generates data for methods that TransCoder handles well. You are right to assume that small methods are more likely to be translated correctly, since there are more opportunities for the model to translate a statement incorrectly in longer methods. However, even for our longest methods (number of tokens > 300), the probability for at least one of the outputs to be correct with beam size 20 is above 10%. The success probability is 47% for our methods with less than 50 tokens, representing 75% of our methods. Surprisingly, very short methods (less than 18 tokens, 25% of our methods) are translated correctly only 34.6% of the time.
>
> > How does it handle test cases with multiple asserts?
>
> EvoSuite is able to generate test cases with multiple asserts. It starts by generating many asserts (most being useless) and then minimizes the output by removing the asserts and test cases that don’t contribute to increasing the criterion. We are able to extract several asserts and rewrite them in other languages.
>
> > I wonder if EvoSuite scaffolding is an issue
>
> The EvoSuite scaffolding is not an issue as it mostly handles things that are common to every test suite (e.g. environment initialization, sandboxing to run unknown code safely). We set up similar environments for other languages, for instance by using firejail to execute code safely.
>
> > When does EvoSuite fail to generate high-quality test suites?
>
> We could see no clear pattern for when EvoSuite manages to generate high quality test suites and we are not aware of any study that would answer this question in the test generation literature.
>
> >Are you generating tests sets for evaluation as well? Does this change the baseline (published) TransCoder computational accuracy numbers?
>
> We do not generate test sets for evaluation as we found it had a very small effect compared to training on the generated parallel data (see Beam reordering in table 6 in the appendix C).
> The authors of TransCoder made some modifications to their tokenization/detokenization code impacting their performances (mostly positively) since the paper was published and they show an updated results table in their github repository. We evaluated their models ourselves and obtained results similar to those they report in their repository. In this paper, we report the maximum between the score we obtained and the score reported in the TransCoder paper.
>
> >Maybe I am missing something, but why would generated test set mismatch to TransCoder tests set be an issue?
>
> This is an issue because we sometimes train our model to generate code that passes the tests generated by EvoSuite but would not pass the tests of TransCoder. It can lead to errors as measured using the TransCoder test set (for instance, our model can be trained to translate the Java int type to np.int(32) in python because it leads to more EvoSuite tests passing but it also leads to a lower score on our test set).
>
> > For multiple published papers the authors cite corresponding arXiv pre-prints.
>
> Thank you, we added references to the journal or conference for several papers in our references, including those you mentioned.
>
> > Minor corrections
>
> Thank you very much. We corrected these sentences

---

### Official Review · Reviewer_19G3 · 2021-11-02

**Correctness:** 3
**Technical Novelty And Significance:** 4
**Empirical Novelty And Significance:** 3
**Recommendation:** 8
**Confidence:** 4

**Main Review:**

The paper is well written, easy to follow and presents an interesting and novel idea. Technically, the work touches many parts - “static” text-to-text transformer, executing code, checking tests passing, observing and optimising test coverage. The paper has quite some effort into it in getting precise results using different decoding strategies and has ablation studies on these factors. What would have been more interesting though is how well would it work with differences in the test generation/acceptance strategies.

The experiment about dropping mutation scores is the most interesting one from this perspective. If I understand correctly, it seems to have no huge effect on the results (other things like the cache are also ablated?). Probably a related question is if computing mutation scores is needed or one can replace it with much cheaper code coverage metrics.

One of the biggest limitations of the approach seems to be around how unit tests are translated between the languages. The paper downplays this step in its expositions by simply saying that simple scripts can translate unit tests. While I agree that unit tests tend to be inputs and outputs for which translation should be easier, this is probably limiting the scope of what can possibly be translated. The authors could include an appendix or discussion on how many of the tests could be translated successfully and what incorrect translations look like - e.g. tests that perform a sequence of APIs or expect exceptions. A related question is if tests could be translated with some of the baseline models such as Transcoder/DOBF and if such an idea was considered and was/wasn’t successful.

It is not completely clear if the data filtering techniques (to only include functions for which tests can be generated) also play a role in the selection of the evaluation data. I would be happy to get a confirmation of this from the authors.


**Summary Of The Paper:**

The paper proposes a technique to generate parallel training data for machine translation between programming languages. This data is generated based on specifics of the programming languages and the availability of programming tools in these languages.

Using this additional data, the paper shows that machine translation using state-of-the-art transformers between programming languages significantly improves over prior data collection methods with back-translation.


**Summary Of The Review:**

The idea of the paper is specific to translation between programming languages and may be of interest to a subset of the ICLR community. There are interesting ideas and overall good evaluation of the paper.

---

> ### Author Response · Authors · 2021-11-16
> **Response to Reviewer 19G3**
>
> >What would have been more interesting though is how well would it work with differences in the test generation/acceptance strategies. The experiment about dropping mutation scores is the most interesting one from this perspective. If I understand correctly, it seems to have no huge effect on the results (other things like the cache are also ablated?).
>
> You are correct, the cache is also ablated in this experiment. It is correct to say that the effect is not huge (we still have clear gains over the baselines when keeping every test) but it is still noticeable (1.4% points). Dropping the mutation scores increases the noise in the inputs of the model (more false negatives) but some of the examples validated by tests with low mutation scores are correct and it seems like there is still enough signal to improve the model overall.
>
> >Probably a related question is if computing mutation scores is needed or one can replace it with much cheaper code coverage metrics.
>
> The mutation score is computed only once when we generate tests for the java functions we selected in our test set. Moreover, computing the mutation score is much less costly computationally than generating the tests. Hence, using code coverage metrics to select our tests would not have a significant impact on the cost of our method. Based on the literature in automated unit test generation, we believe that this metric better correlates with the quality of the tests than coverage metrics and that it is worth computing.
> We also add the mutation score in the criterion optimized by EvoSuite and observed that it resulted in better mutation scores with a constant time budget of 20 seconds. We could consider lowering the time budget and dropping the mutation score from the criterion to enable more computational-intensive methods to generate parallel data (e.g. generating tests from java translations of python functions).
>
> >One of the biggest limitations of the approach seems to be around how unit tests are translated between the languages. The paper downplays this step in its expositions by simply saying that simple scripts can translate unit tests. While I agree that unit tests tend to be inputs and outputs for which translation should be easier, this is probably limiting the scope of what can possibly be translated. The authors could include an appendix or discussion on how many of the tests could be translated successfully and what incorrect translations look like - e.g. tests that perform a sequence of APIs or expect exceptions.
>
> You are right. We indeed acknowledge that translating arbitrary unit tests would be difficult and this is a limitation of our method. As you mentioned, translating tests expecting exceptions would be particularly difficult as there is often no clear mapping between exceptions in different programming languages. About 5.6% of the test suites we generated expected an exception in at least one test case, and less than 2% among the test suites with high mutation scores. In those cases, the translation fails and no parallel examples can be generated. We added a paragraph in the appendix to discuss this subject. Thank you for mentioning this.
>
>
> >A related question is if tests could be translated with some of the baseline models such as Transcoder/DOBF and if such an idea was considered and was/wasn’t successful.
>
> We did not try translating the unit tests with a machine learning model in this work. This choice is linked to the test creation framework we selected (i.e. EvoSuite). With the tests it generates, it is very easy to extract their semantics and translate them quickly and reliably using simple regular expressions. However, we believe that this approach could be used to generalize our approach to more types of unit tests for which writing a rule-based translator would be difficult (e.g. tests created by humans and extracted from the code or created with neural tools such as AthenaTest). Wrong translations could cause some false positives and we would want to ensure that incorrect translations are unlikely to pass incorrectly translated tests.
>
> > It is not completely clear if the data filtering techniques (to only include functions for which tests can be generated) also play a role in the selection of the evaluation data. I would be happy to get a confirmation of this from the authors.
>
> We can confirm that our data filtering techniques are *not* applied on the validation and test sets. We keep the whole unfiltered test set from TransCoder and we cannot create high-quality test suites for many of the elements in the test set. We try to clarify this in the Evaluation subsection by adding the adjective “full” for the validation and test sets of TransCoder. Thank you for pointing this out!

---

### Official Review · Reviewer_zd7L · 2021-11-04

**Correctness:** 4
**Technical Novelty And Significance:** 2
**Empirical Novelty And Significance:** 2
**Recommendation:** 5
**Confidence:** 4

**Main Review:**

**Strengths:**
- A simple method is proposed to fine-tune existing code translation models without the need to manually curate parallel data
- The method does not require any changes to the training procedure of the model, and can be independently applied in the fine-tuning stage
- The method can potentially be applied to fine-tune a model for characteristics other than correctness. for ex, runtime efficiency, code style, etc.
- Results show significant improvement over the baseline TransCoder and DOBF models


**Weaknesses:**
- If ground truth parallel data were available for this task, then fine-tuning a model on this data would naturally result in gains in performance. Since this parallel data is extremely scarce, the authors propose to utilize an automated test generation suite to build this parallel data from the model translations itself. I have 2 concerns with this proposal:
   1. It requires an automated unit test generation tool to be available for at least one of the languages translated. Unit tests generation is an active area of research and tools are available only for limited sets of languages and primarily for modern languages such as Java, Python, etc. An important premise of automated code translation -- which the authors mention in their introduction and ethical concerns section -- is about modernizing legacy languages such as COBOL, where automated test generation systems might not be available at all. The proposed method therefore, in my opinion, is bottlenecked on the availability of automated unit test generation tools, and is not applicable when they are not available.
   2. Automated unit test generation tools can themselves generate incorrect test cases -- test cases testing the incorrect method, incorrectly testing the method resulting in false positives or false negatives. AthenaTest [1], for instance, works better than EvoSuite and generates correct unit tests for the intended method only 16.21% of the time as per [1]. 26.71% of the times the test case generated by AthenaTest test for the wrong behavior. Other factors such as dynamic typing (as in Python) [2] or the use of more complicated data structures (such as Generics in Java [3]) further affects the effectiveness of these test generation systems. It would have been nice to get some insights into the quality of test cases generated and how this affects the performance of the translation model. This effect, I believe, would be especially pronounced in cases where the code translated is a piece of more complicated codebase and not standalone functions that can be compiled in isolation.

- The authors utilize EvoSuite more as a test data generation tool than a test case generation tool. EvoSuite generates test cases for Java, while this work trains models for Java, Python and C++. Therefore, if I understand correctly, the way test cases are created for Python and C++ is by using the expected input/output pairs and assert statements generated for corresponding Java code, and putting these in a template for the other languages. I'm not completely convinced that this is an ideal strategy that can generalize to a broader set of languages or for more complicated pieces of code. For example: if the source code implements a Java class and the generated test operates on this class and their methods, how would this be translated to Python test without changing the procedure for Python test case generation? Or, how would this procedure work when there isn't a direct mapping of data types and constructs between the languages. This further reduces the applicability of this approach.

- If I understand correctly, the evaluation used in this work includes a reference implementation and a translated implementation, which are tested to generate the same output given the same input? This creates problems as authors note in Figure 4, where the gold translation is equivalent to the source code only on a small domain, while the model generated translation, though correct and better is incorrectly marked wrong. This, to me, indicates an issue with the utilized test set where gold implementations might not be correct translations, thereby affecting the scores. Contemporary works in code generation and translation [4,5] have utilized expected input / output pairs to test the performance of the code generation system, and I wonder why the authors here chose to compare the performance against a reference implementation and not against input-output pairs directly. Additionally, we don't know how prevalent the problem of incorrect reference translations is in the utilized test set.


**Other concerns/Suggestions:**
- The authors mention that they collected data from GitHub with permissive licenses. Can the authors please share the list of licenses used for this data collection step? Having this information in the appendix will provide a reference point for future work in this space on which licenses might be acceptable to use when working with GitHub data.


[1] Tufano, Michele, et al. "Unit Test Case Generation with Transformers." arXiv preprint arXiv:2009.05617 (2020).

[2] Lukasczyk, Stephan, Florian Kroiß, and Gordon Fraser. "Automated Unit Test Generation for Python." International Symposium on Search Based Software Engineering. Springer, Cham, 2020.

[3] Vogl, Sebastian, et al. "EVOSUITE at the SBST 2021 Tool Competition." 2021 IEEE/ACM 14th International Workshop on Search-Based Software Testing (SBST). IEEE, 2021.

[4] Hendrycks, Dan, et al. "Measuring Coding Challenge Competence With APPS." arXiv preprint arXiv:2105.09938 (2021).

[5] Chen, Mark, et al. "Evaluating large language models trained on code." arXiv preprint arXiv:2107.03374 (2021).

**Summary Of The Paper:**

- This work presents a way to fine-tune code translation models by providing weak indirect supervision from automated test case generation tools
- The proposed approach has 3 main components
   - Pre-trained models for code translation: the authors use Transcoder and DOBF for this.
   - Automated test case generation tool: the authors use EvoSuite, an evolutionary computing based automated test case generation tool for Java, and
   - Dataset: the authors collect approximately 330k Java code samples from Github with permissive licenses
- The proposed approach then proceeds as follows:
   - It starts by creating unit tests for Java code samples using EvoSuite
   - It uses the generated expected input / output pairs to create unit tests for other languages -- Python and C++
   - The method then translates code samples from one language to another and then validates the generated translation using the unit tests created in the step above
   - In this manner, the approach collects parallel translation data samples that are later used for fine-tuning the model
- The authors propose 2 variants of the method: offline and online
   - In the offline variant, the approach utilizes all the parallel code samples collected to fine-tune the model, while
   - In the online variant, the approach maintains a cache which is updated with new examples generated from the model
- Models fine-tuned on parallel data created in this manner show substantial increase in performance over TransCoder and DOBF models.

**Summary Of The Review:**

The proposed method achieve significant improvements over the baseline TransCoder and the DOBF models. However, I found the following weaknesses, resulting in my recommendation.

1. Bottleneck on the availability of test generation tools thus limiting the number of languages it can be applied to.
2. Quality of generated tests dips on more complicated code segments, and we do not know what effect that has on the performance. The method is trained on Java functions that can be executed in isolation.
3. Method to generate test cases for translated Python and C++ codes seems to be dependent on expected input/output pairs, and it's not clear how the proposed method would work for more complicated code segments, or for languages with no direct mapping of data types and constructs.

---

> ### Author Response · Authors · 2021-11-16
> **Response to Reviewer zd7L 1/2**
>
> >It requires an automated unit test generation tool to be available for at least one of the languages translated. Unit tests generation is an active area of research and tools are available only for limited sets of languages and primarily for modern languages such as Java, Python, etc. An important premise of automated code translation -- which the authors mention in their introduction and ethical concerns section -- is about modernizing legacy languages such as COBOL, where automated test generation systems might not be available at all. The proposed method therefore, in my opinion, is bottlenecked on the availability of automated unit test generation tools, and is not applicable when they are not available.
>
> We agree that our method depends on the existence and quality of unit tests generation tools. However, we believe that our method can be used even if no such tool is available for the languages of interest for the following reason:
> Java can be added as one of the languages to translate. For instance, in this paper, we use no unit test creation tool for either C++ or Python but our results show that we can still significantly improve the performance of TransCoder for C++ -> Python or Python -> C++ translation. For COBOL to C++ translation, one could train a model on C++, COBOL and Java and create multilingual unit tests from Java code. Creating and using a test creation tool for COBOL or C++ would probably increase the size of the parallel dataset created (because only one translation needs to be correct) and improve the performance of the model but it is not necessary.
>
> > Automated unit test generation tools can themselves generate incorrect test cases -- test cases testing the incorrect method, incorrectly testing the method resulting in false positives or false negatives. AthenaTest [1], for instance, works better than EvoSuite and generates correct unit tests for the intended method only 16.21% of the time as per [1]. 26.71% of the times the test case generated by AthenaTest test for the wrong behavior.
>
>
> As you mentioned, all test generation systems can generate incorrect tests. For example, test cases that test for the wrong behaviour, don't compile, or cause runtime errors. Fortunately, our approach would not be compromised by this, because it's easy to filter out all such incorrect tests and both EvoSuite and AthenaTest return test suites with only correct test cases.
>
> Nevertheless, we would of course, prefer to have the maximum number of correct tests available to the technique in order to limit the number of false positives. As you say AthenaTest may have the edge over EvoSuite in this regard.
>
> More generally, the particular test generation technique can be seen as a parameter in our overall approach, and so we could use AthenaTest, or some other technique in place of EvoSuite.  We have added a clarification of this in the revised version of the paper.
>
> Our primary motivation for using EvoSuite is that it is an open source, widely used by testing researchers, and therefore it would be easy for them to replicate our results. However it is a choice, and others are also free to substitute in different test generation techniques as a component to overall approach.
>
>
> > It would have been nice to get some insights into the quality of test cases generated and how this affects the performance of the translation model.
>
> We use the mutation score to evaluate the quality of the tests and added a histogram of the mutation scores we obtained in the appendix. In our ablation study, we included an ablation where we keep the tests regardless of the mutation score and show that it decreases the average performance by 1.4% points. Early experiments showed us that the performance of the model varies little with small changes of the mutation score threshold (which matched our expectations after seeing the distribution of the mutation scores).

---

> > ### Author Response · Authors · 2021-11-16
> > **Response to Reviewer zd7L 2/2**
> >
> > >If the source code implements a Java class and the generated test operates on this class and their methods, how would this be translated to Python test without changing the procedure for Python test case generation? Or, how would this procedure work when there isn't a direct mapping of data types and constructs between the languages. This further reduces the applicability of this approach.
> >
> > We agree that this is a limitation of our current approach. We believe that our approach is useful for cases for which we do not generate parallel unit tests (e.g. functions returning user-created classes) because some patterns we learn using the unit tests (e.g. choosing between the / and // operators correctly or correctly accounting for operator precedence as shown in Figure 1) would still be present in these functions. Unfortunately, building a test set to measure how much improvement our method yields in those cases would be more difficult than for standalone functions.
> > We also believe that our method could be extended to a much wider domain of functions and methods if it was used to translate larger scopes of code (e.g. whole files or functions plus all their dependencies).
> >
> > > If I understand correctly, the evaluation used in this work includes a reference implementation and a translated implementation, which are tested to generate the same output given the same input? This creates problems as authors note in Figure 4, where the gold translation is equivalent to the source code only on a small domain, while the model generated translation, though correct and better is incorrectly marked wrong. This, to me, indicates an issue with the utilized test set where gold implementations might not be correct translations, thereby affecting the scores. Contemporary works in code generation and translation [4,5] have utilized expected input / output pairs to test the performance of the code generation system, and I wonder why the authors here chose to compare the performance against a reference implementation and not against input-output pairs directly. Additionally, we don't know how prevalent the problem of incorrect reference translations is in the utilized test set.
> >
> > We agree that comparing the results on lists of input-output pairs would be more appropriate to ensure that the input and output functions are equivalent. However, for some use cases such as proposing translations to help students learn about new programming languages, returning the more pythonic version of factorial using the python int type would arguably be better than using the np.int32 when translating the java function shown in Figure 4. It is especially true if, like in this case, this function is not intended to be run on inputs causing overflows. In this work, we did not build a new validation/test set and used the one created by the authors of TransCoder. It allows us to compare ourselves easily to previous works (e.g. TransCoder and DOBF) using the same dataset. The contemporary works you cited ([4,5]) study code generation from natural language prompts and check the validity of the generated code using a series of unit tests. Therefore, they do not provide datasets for translation between programming languages. We believe that the test dataset of TransCoder is similar to theirs as it checks that the output of the generated translation is the same as that of the ground truth on a series of unit tests, but we agree that creating the input/output pairs based on the input function would also make sense in the context of code translation.
> >
> > > The authors mention that they collected data from GitHub with permissive licenses. Can the authors please share the list of licenses used for this data collection step? Having this information in the appendix will provide a reference point for future work in this space on which licenses might be acceptable to use when working with GitHub data.
> >
> > Thank you for pointing this out. We added the list of licenses we keep when the downloading the dataset from Google BigQuery in a footnote: 'apache-2.0', 'mit', 'gpl-2.0', 'gpl-3.0', 'bsd-2-clause', 'bsd-3-clause'.

---

### Official Review · Reviewer_a7Vt · 2021-11-04

**Correctness:** 1
**Technical Novelty And Significance:** 2
**Empirical Novelty And Significance:** 2
**Recommendation:** 6
**Confidence:** 3

**Main Review:**

Overall, I like this paper. The idea of using unit tests for transpilation is, although not formally guaranteed, a reasonable approach for transpilation in my opinion.

Some of the language in the paper is a little off-putting as it seems to be disconnected (in my opinion) from deeply understanding the space of natural language translation (unstructured ambiguous languages) vs. programming language translation (structured unambiguous languages). It's my hope that if this paper is accepted, that the authors will polish the writing slightly.

For example: "translation systems are not as effective for source code as for natural languages." This is a weird claim. The work that Alvin Cheung (Berkeley) and JRK (MIT) for Maaz Ahmad (UW Washington / Adobe Research) is wildly effective and has automatically transpiled ~300 functions with a geometric mean performance improvement of 3.36x. Moreover, this approach is *formally verified* unlike the authors approach. In addition, it's already being used in production quality code (Adobe Photoshop v.21).

There are many other such examples that seem to demonstrate the authors perhaps need to do a bit deeper literature review before making such bizarre claims. For example, the word "transpilation" never even appears in the paper, yet the whole paper is about transpilation. Do the authors not know about the field of "transpilation" and that they are working in this space?

https://en.wikipedia.org/wiki/Source-to-source_compiler

These minor issues aside, I actually like the approach and am favorable toward the paper.


**Summary Of The Paper:**

This paper is aimed at using the automatic generation of unit tests as a means to perform transpilation (i.e., the translation of code from one language to another). The authors demonstrate their technique from Java to Python and from Python to C++. They claim that their approach outperforms prior methods by +16% and +24% over prior state-of-the-art.


**Summary Of The Review:**

TL;LD: it's an important emerging area. It's a reasonable (although not formal) way to perform transpilation. Using self-supervision is likely one of the most important aspects of the system because 99% of the code that we have today is currently unlabeled. To my knowledge, only a handful of labeled datasets exist (e.g., POJ-104, Google Code Jam, IBM/MIT's Project CodeNet, Microsoft's CodeXGLUE, etc.)

Weak accept.

---

> ### Author Response · Authors · 2021-11-16
> **Response to reviewer a7Vt**
>
> >Some of the language in the paper is a little off-putting as it seems to be disconnected (in my opinion) from deeply understanding the space of natural language translation (unstructured ambiguous languages) vs. programming language translation (structured unambiguous languages). It's my hope that if this paper is accepted, that the authors will polish the writing slightly.
>
> Thank you. In the updated version of the paper, we tried to do a better job of explaining the more general differences between natural and programming languages and added more information about transpilation in other contexts. For instance, we added references to verified lifting in the introduction and added some examples of language pairs for which transpilers are very effective (e.g. Java to Scala).
> We thank you for your remarks, which allowed us to improve the quality of the paper.
>
> > For example: "translation systems are not as effective for source code as for natural languages." This is a weird claim. The work that Alvin Cheung (Berkeley) and JRK (MIT) for Maaz Ahmad (UW Washington / Adobe Research) is wildly effective and has automatically transpiled ~300 functions with a geometric mean performance improvement of 3.36x. Moreover, this approach is formally verified unlike the authors approach. In addition, it's already being used in production quality code (Adobe Photoshop v.21).
>
> We agree that this sentence is misleading and sounds more general than we intended. It is true that very efficient automatic translation methods exist for some programming language pairs (e.g. Java to Scala). Moreover, as you said, some transpilers are formally verified (or are at least non-stochastic), which is a clear advantage over state-of-the-art translators in the field of NLP. We modified this claim to more clearly explain that it applies only to translation *between arbitrary programming languages* and added further references about those language pairs for which automatic translation tools are known to be efficient and/or formally verified. Thank you for noticing and mentioning this issue, thereby helping us to clarify and sharpen our claims.
>
> > For example, the word "transpilation" never even appears in the paper, yet the whole paper is about transpilation. Do the authors not know about the field of "transpilation" and that they are working in this space?
>
> We agree with you and your comment shows us that using the term “transpilation” would probably help some readers discover more papers on this subject, or help researchers with a background in transpilation to find our paper. We added this term and more context in the introduction. Thank you for your comment.
>
> > TL;LD: it's an important emerging area. It's a reasonable (although not formal) way to perform transpilation. Using self-supervision is likely one of the most important aspects of the system because 99% of the code that we have today is currently unlabeled. To my knowledge, only a handful of labeled datasets exist (e.g., POJ-104, Google Code Jam, IBM/MIT's Project CodeNet, Microsoft's CodeXGLUE, etc.)
>
> We agree. Our main motivation for developing an unsupervised method for code translation was that existing parallel datasets for code translation are limited to a few language pairs (e.g. translation tasks for CodeXGLUE) or in a very specific domain (i.e. coding competitions).
>
>
> > 1: The main claims of the paper are incorrect or not at all supported by theory or empirical results.
>
> We agree that some sentences were poorly formulated, and hope that we have addressed them in the updated version of the paper. Thank you for helping us to improve the paper, in particular our poor formulations which seemed much broader than we intended. Please let us know if you still have any doubt about the validity of our claims.

---

### Author Response · Authors · 2021-11-16
**General response to reviewers**

We thank all the reviewers for their insightful feedback, which allowed us to improve the paper. We replied to each reviewer individually and updated the paper following their suggestions. Here are the main modifications we made:
- We added more context about transpilation in the introduction, by citing relevant related works from this domain. We modified our sentence, which was comparing source code translation to translation in the context of NLP, to make it much less general and align it with what we actually meant (i.e. that translation between arbitrary languages is still an open problem and not deployed and used as widely as in the context of NLP).
- We added a footnote with the full list of licenses we use to filter the repositories available on Google BigQuery.
- We added a sentence in the conclusion to highlight that our method significantly improved the performance of models translating between C++ and Python even though no test generation tool was used for either of these languages (but exclusively for java).
- We added a histogram of the mutation scores of our tests in the appendix.
- In the Evaluation section, we now explain that we evaluate our models on the *full* test sets of TransCoder to clarify that we do not filter the test dataset of TransCoder based on the mutation score of the unit tests or anything else.
- We added a paragraph with more information about what happens in case of unit tests translation failure, and how frequently it happens due to test cases expecting exceptions.
- In the related work section on unit tests generation, we added a sentence explaining that other test frameworks could be substituted or used in addition to EvoSuite.
- We corrected the poorly formulated sentences highlighted by reviewer LnDd and added a reference to the conference instead of to the arxiv preprint when citing published papers.

---

### Decision · Program_Chairs · 2022-01-20

**Decision:**

Accept (Spotlight)

**Comment:**

This paper is about unsupervised translation between programming languages. The main positive is that it introduces the idea of using a form of unit test generation and execution behavior within a programming language back-translation setup, and it puts together together a number of pieces in an interesting way: text-to-text transformers, unit test generation, execution and code coverage. Results show a substantial improvement. The main weaknesses are that there are some caveats that need to be made, such as the (heuristic, not learned) way that test cases are translated across languages is not fully general, and that limits the applicability. There are also some cases where I find that the authors are stretching claims a bit beyond what experiments support, e.g., in the response to zd7L about applicability to COBOL.

All-in-all, though, it's a good implementation of an idea that should have a lasting place in this line of work, so it's worth accepting.